# A Study of Leisure Constraints and Job Satisfaction of Middle-Aged and Elderly Health Care Workers in COVID-19 Environment

**DOI:** 10.3390/healthcare9060713

**Published:** 2021-06-10

**Authors:** Chien-Hung Wu, Hsiao-Hsien Lin, Sin-Yu Lai, Kuan-Chieh Tseng, Chin-Hsien Hsu

**Affiliations:** 1Department of Marine Recreational, National Penghu University of Science and Technology, Penghu County 880011, Taiwan; wu1023@gms.npu.edu.tw; 2Department of Leisure Industry Management, National Chin-Yi University of Technology, Taichung 41170, Taiwan; chrishome12001@yahoo.com.tw (H.-H.L.); love917526@gmail.com (S.-Y.L.); 3MA Program in Social Enterprise and Cultural Innovation Studies, College of Humanities & Social Sciences, Providence University, Taichung 41170, Taiwan; jackt72@pu.edu.tw

**Keywords:** discussing pressure, on-duty mechanism, motivation, friendly workplace environment, high-level medical personnel

## Abstract

The purpose of the study was to examine the leisure constraints and job satisfaction of middle-aged and elderly health care workers. The study employed a mixed research method, utilizing SPSS 22.0 and AMOS 23.0 statistical software to analyze 260 questionnaires using basic statistical tests, *t*-tests, ANOVA tests, and structural equation models, and then interviewed medical and public health workers and experts in the field, and the results were analyzed using multivariate verification analysis. The results showed that there was a significant low correlation between leisure constraints and job satisfaction among middle-aged and elderly health care workers (*p* < 0.01); interpersonal constraints and external job satisfaction factors were the main influencing factors; improving promotion opportunities and receiving appreciation increased job satisfaction; poor working environment and facilities, as well as the lack of achievement, were the main factors that reduced satisfaction; health factors, a lack of family support, no exercise partner, and a lack of extra budget are the key to leisure constraints. If the organization can provide nearby sports facilities for middle and high-age medical workers, improve welfare, and increase willingness to participate in leisure activities, physical and mental health can be improved. Finally, interpersonal interaction in leisure obstacles is the main reason for improving job satisfaction.

## 1. Introduction

In the severe COVID-19 environment, countries around the world are facing serious challenges to their overall economic, social, and environmental development, due to the uncertain infection risk in their living environment, which has led to overall social unrest and has had an impact on industrial development [1,2,3,4]. In order to strengthen national security, governments have adopted quarantine policies, planned epidemic prevention measures, monitored the physical and mental health of their populations, and provided facilities and human resources to treat confirmed patients and assist people with nucleic acid testing to strengthen the national epidemic prevention frontline. However, the epidemic has not yet been alleviated. Although vaccines have been successfully developed, they are not yet widely available and effective [5,6,7]. The cumulative number of confirmed cases has reached 106 million, with more than 2.31 million deaths [8], and the epidemic has not yet been officially alleviated. Since the outbreak, not only has the epidemic not improved, but mutations of the virus have also emerged [9], continuing to impact the development of governments and interfere with the life quality of the population. For health care workers, treating patients with COVID-19 and other variants of the virus, in addition to the existing medical care, has greatly increased their burden and pressure.

The health of medical personnel is crucial to the effectiveness and quality of medical care [10]. Although young people are the main source of manpower in the workplace [11], middle- and high-level workers play an important role in the command and control of the medical field [12]. In particular, middle-aged and elderly health care workers have accumulated years of health care expertise and have perfect practical experience [13,14], which is an important cornerstone for constructing a sound medical defense system and service quality. However, the daily working hours of medical workers can be divided into morning, afternoon, evening, and large inter-day night shifts. The shift duration is 8–10 h per shift, often up to 12 h when necessary. The duty schedule also requires weekly rotations [15]. Plus, the pressure of the COVID-19 epidemic and the existing medical patients has impacted the original medical work pressure [16], resulting in a health crisis among medical personnel [17] and the occurrence of burnout or resignation [18,19,20,21,22]. In particular, middle-aged and elderly individuals do not have sufficient leisure time, due to work and life pressures [23], and the job rotation mechanism and job responsibilities of health care workers impose tremendous occupational stress [24], resulting in health care workers suffering from burnout and even leaving their jobs, which will affect the overall health care quality and epidemic prevention effectiveness of the country [25,26,27]. Therefore, we believed that helping health care workers to find ways to resist stress in the COVID-19 environment, to preserve the physical and mental health of health care workers, to maintain the employment willingness of health care workers, to stabilize workplace manpower, and to uphold health care quality are important research issues at present [28,29].

Job satisfaction is the key to stabilize the employment willingness of workers, and job satisfaction can be explored in terms of personal factors and work environment. A good working environment can make employees feel more comfortable at work, and this process can be considered as an external non-reward motivational incentive [30]. An individual’s job satisfaction is based on the extent to which the work environment meets the individual’s needs and the individual’s ability to cope with the job requirements [31]. Income, job characteristics, autonomy, stability, respectability, contribution, supportive attitude of supervisors, and better working conditions all have direct positive effects on job satisfaction [32], among which job content, salary, promotion, relationship with supervisors and peers are the main key influences on job satisfaction [33,34]. The higher the level of fatigue, the lower the job satisfaction [35], and the higher the job stress, the lower the job satisfaction or organizational commitment, and the higher the tendency to leave [36]. Therefore, we believed that improving the physical and mental health of the middle-aged and elderly health care workers would be the key to increasing work engagement and concentration, enhancing job satisfaction, and improving job performance and quality of care.

Engaging in leisure exercise can improve personal health [37], develop good leisure exercise habits, relieve psychological stress, enhance physical and mental health [38,39], maintain work performance and quality, and ensure job satisfaction of employees [40]. However, participation in leisure activities is critically influenced by the internal and external environment. Middle-aged and elderly health care workers may not develop regular exercise habits [16,23,24], due to spatial and physical constraints [41], as well as internal factors such as location, time, and cost [42], thus creating obstacles for leisure participation. The obstacles to leisure participation can also be explored in terms of personal, psychological, and time factors, as well as a lack of knowledge, facilities, accessibility, companions, and interests. Leisure constraints also affect the frequency of participation in leisure activities [43,44]. Although previous studies have suggested that removing leisure obstacles can lead to consistent participation in leisure sports and greater job satisfaction [45,46], given the high influence of middle-aged and elderly healthcare workers in upholding the healthcare system [13,14], it may not be easy to completely eliminate leisure constraints, improve the physical and psychological stress of middle-aged and elderly healthcare workers caused by the pressure of work and social environments, reduce the willingness to leave, and maintain the quality of healthcare [47,48].

In conclusion, people’s inability to engage in leisure activities may result in ineffective relief of personal stress [37]. Empirical evidence suggests that increased stress affects people’s life and work performance [43,44]. The failure to achieve job satisfaction tends to make companies dissatisfied with their employees [32], affects employees’ emotions at work [36], accelerates the decline in job performance, and affects employees’ willingness to stay in their jobs. Thus, there is a significant correlation effect between leisure constraints and job satisfaction [45,46]. In addition, after reviewing the literature on COVID-19, medical personnel, physical and mental health, leisure constraints, and job satisfaction, it was found that medical devices and facilities [45], the physical and mental health of doctors and nurses [28,29,49,50,51], followed by physical and mental health and job satisfaction [49], were the most frequently studied issues related to medical personnel in the COVID-19 environment. Fewer researchers have explored job satisfaction [52], and almost no studies have examined the relationship between leisure constraints and job satisfaction.

Therefore, this study was conducted to identify the most important factors between leisure constraints and job satisfaction among middle-aged and elderly health care workers, so that the smallest magnitude of adjustment in leisure behaviors could lead to the largest increase in job satisfaction, and suggestions for improving the current leisure activities of middle-aged and elderly health care workers can be made. The major goals of this study are to help health care workers find ways to resist stress in the COVID-19 environment, maintain the physical and mental health of health care workers, maintain the employment willingness of health care workers, stabilize workplace manpower, and uphold the quality of health care.

## 2. Literature Discussion

### 2.1. Leisure Constraints

The perception of dislike when an individual is prevented from participating or engaging in leisure activities by one or more factors [53], especially in the COVID-19 environment, where the threat of an epidemic makes it impossible to regularly engage in any leisure activities and recreation [54] and creates a negative feeling when people cannot participate or engage in leisure activities, can be considered as leisure constraints [55].

Leisure constraints can produce internal, interpersonal, and structural obstacles [56]. Spatial obstacles, physical obstacles [41], and intrinsic factors, such as location, time, and cost [42], are the main keys to not developing regular exercise habits [16,23,24], while learning institutions, psychological factors, and curricular factors are leisure constraints in middle-aged and elderly populations [57].

Leisure activities can improve the physical and mental health of individuals and maintain a good physical and mental state [38,39]. However, under the threat of the COVID-19 epidemic, people feel alienated and worried about their surroundings, which leads to changes in lifestyle habits [58]. The lower immunity of middle-aged and elderly adults [57], the high intensity of medical work, and the requirement to comply with professional ethics [23,24,25] increase the barriers for health care workers to participate in leisure activities. Therefore, exploring leisure constraints can help identify the factors that prevent middle-aged and elderly health care workers from engaging in leisure activities and identify ways to promote leisure activities to maintain health.

### 2.2. Job Satisfaction

Job satisfaction is a positive affective orientation [59,60,61] that occurs when individuals have a pleasant and positive emotional state [60] about their work or work experience in terms of their workplace or work content. Job satisfaction is very important for health care workers, especially in the COVID-19 environment, and good job satisfaction affects their performance and is the key to the quality of health care and the effectiveness of epidemic prevention [47].

Job satisfaction involves comparing the expected job content and rewards with the actual ones in a specific environment, and positive satisfaction occurs when the actual rewards or feelings received are higher than expected; conversely, dissatisfaction occurs [62]. Job satisfaction can be considered in terms of personal emotions, work environment, benefits, and organizational expectations [63], where the facility environment, salary, promotion opportunities, supervisor recognition, self-actualization, gaining a sense of accomplishment, and personal recognition of the job and corporate values [64,65,66,67] are the main subtle keys.

High-quality job satisfaction is the key to maintaining employee performance and the quality of corporate services [68]. In particular, the rising epidemic of COVID-19 has led to soaring stress in the healthcare environment, which has exacerbated the physical and mental health of middle-aged and elderly healthcare workers, and poor physical and mental health status plays a major role in influencing individual performance and the overall quality of healthcare [35,36]. Therefore, it is the second focus of this study to identify the inadequacies of middle-aged and elderly health care workers in the workplace to help improve the working environment and conditions to meet their workplace needs, improve their work performance, and maintain the quality of health care.

### 2.3. Relevant Research on Leisure Constraints and Job Satisfaction

Advances in technology have raised the standard of medical care. However, there are still many different types of illnesses and frequent accidents and injuries. Medical workers have a great responsibility to treat and restore people’s health. The combination of heavy workload and aging increases the stress of middle-aged and elderly health care workers [24], making them feel rejected by the content of their work [37,43,44] and decreasing their willingness to work [45,46].

Research confirms that leisure exercise can effectively help employees relieve stress, improve physical and mental health, maintain work performance, and maintain stable job satisfaction [37,38,39,40]. Enhancing personal commitment to leisure activities and lowering leisure constraints can improve the feelings of investment in activities or work [69], improve employees’ leisure experiences, reduce the conditions that prevent them from engaging in leisure, and help alleviate the negative emotions of work restrictions arising from the work process [70]. In particular, in the current severe medical and epidemic prevention environment, the work pressure of healthcare workers is gradually increasing. Compared with younger workers, middle-aged and elderly healthcare workers lack opportunities to relieve stress, due to work, time, and environmental factors [41,42], and their mental and physical health defense mechanisms are deteriorating [59].

Moreover, although there is a wide range of factors that affect job satisfaction [32], the level of stress during work is the key to job satisfaction or organizational commitment [35,36]. An enthusiastic work attitude helps to maintain positive employee behavior, improve concentration, and further stabilize work performance [71], and one of the best ways to maintain personal enthusiasm, such as work or activities, is to keep one’s physical and mental health [38].

According to the above literature, the work of healthcare professionals is delicate and complex, and carries a high degree of risk and responsibility. For middle-aged and elderly health care workers, the pressure further increases as they age. Without proper adjustment to relieve work pressure over time, the effectiveness of work will be affected. This can lead to frustration, increased stress, and decreased willingness to practice. Therefore, the present study believes that understanding the main causes of leisure constraints among middle-aged and elderly health care workers, reducing the conditions of leisure constraints, and improving leisure awareness and behavior will help to relieve work stress, mediate personal emotions, improve job satisfaction, and ultimately increase their desire to stay. Therefore, this study aimed to identify the main key factors that influence the relationship between leisure constraints and job satisfaction, and to help middle-aged and elderly health care workers to improve their physical and mental health, enhance their job performance, obtain the greatest job satisfaction and stabilize the quality of health care with the least adjustment of leisure activities, which will be the main focus of this study.

There are few studies on leisure constraints and job satisfaction among middle-aged and elderly health care workers. Nevertheless, it is believed that physical and mental health can be maintained by engaging in leisure activities [71], and that good physical and mental health affects the efficiency of life or work [38]. Work efficiency affects the evaluation of employee performance by supervisors in the workplace [35,36]. In order to improve the evaluation of the workplace, employees become more engaged in their work and even feel burned out [70], which eventually can affect job satisfaction [69].

The data composition was observed using measurement and structural models from the structural equation modeling theory. Confirmatory factor analysis (CFA) [72] and path analysis were used to investigate the relationship between potential variables [73]. We believe that the structural equation modeling theory can be used to obtain answers to the effects related to the leisure constraints and job satisfaction of middle-aged and elderly health care workers.

## 3. Methodology

### 3.1. Study Design, Population, and Setting

The study analyzed the current status of leisure constraints and job satisfaction among middle-aged and elderly health care workers by investigating the effects between leisure constraints and job satisfaction, and attempted to identify the smallest changes in leisure behaviors to obtain the largest improvements in physical and mental health in the workplace, as shown in Figure 1.

Based on the above framework, the researcher has produced three hypotheses in total.

**Hypotheses** **1** **(H1).**
*Assume that the middle-aged and elderly healthcare workers have the same view on leisure barriers.*


**Hypotheses** **2** **(H2).***Assume that the middle-aged and elderly healthcare workers have the same view on job satisfaction*.

**Hypotheses** **3** **(H3).**
*Assume*
*that*
*leisure barriers are significantly related to job satisfaction.*


According to the literature mentioned above, engaging in leisure activities maintains a state of physical and mental health [69], and good physical and mental health affects productivity [38]. If employees are able to improve their productivity, it will influence the performance evaluation of employees by workplace supervisors [35,36]. Personal workplace evaluations will change employees’ attitudes and perceptions toward their jobs [71] and ultimately affect job satisfaction [70]. Therefore, we believe that there may be a mutual effect between leisure constraints and job satisfaction of middle-aged and elderly health care workers. Therefore, it was hypothesized that there would be a significant effect between leisure constraints and job satisfaction.

While scientific research needs to be conducted with adequate theoretical support, rarity or novel research areas are relatively weak in the theoretical foundation. Adopting a complex research approach, complementing the breadth of research with quantitative research methods [70,71], and deepening the depth of research with qualitative research [72] can compensate for research methodological or theoretical shortcomings [73]. The study adopted a mixed research method. First, a sample of middle-aged health care workers from the medical workers at the Central Hospital of Taiwan was created using deliberate sampling and an online questionnaire survey platform. Then, 300 questionnaires were distributed using snowball sampling and 260 valid questionnaires were recovered, with an effective questionnaire rate of 87%. If the sample can obtain 5 multiples of the variable [74,75], or if the number of questionnaires exceeds 100 [75,76,77], it was considered representative. Using SPSS 22.0 and AMOS 23.0 statistical software, we examined the current status of leisure constraints and job satisfaction among middle-aged and elderly health care workers using basic statistics, and then analyzed the relationship between the two using a structural equation model. Then, on-the-job health care workers or scholars in the fields of health care, public health, or human resource management were interviewed to provide insights based on the analysis results, and, finally, the information was compiled, organized, and analyzed in order to construct the report [78]. Finally, the multivariate verification analysis method was used to integrate the information of different research subjects, research theories and methods, and to obtain accurate knowledge and meanings by comparing the research results from multiple perspectives and multiple data [79,80].

### 3.2. Measurements

The questionnaire adopted a 5-point Likert scale, with a score of 1 being very dissatisfied and 5 being very satisfied. After the content was edited with reference to the literature, three experts were sought to examine the content, and then SPSS 22.0 statistical software was used to determine the topic and then test statistically. A Kaiser–Meyer–Olkin (KMO) value > 0.06 and a *p*-value of less than 0.01 (*p* < 0.01) in Bartlett’s test indicated that the scale was suitable for continuous factor analysis [81]. A coefficient α greater than 0.60 indicated that the questionnaire had good reliability [82].

In the questionnaire, the background of the respondents was composed of gender, 45–64 years old age group, unmarried, married and other (divorced) marital status, junior college, university or postgraduate education. 

The questionnaire refers to related literature on leisure barriers [23,24,25,38,39,58,59] and job satisfaction [47,60,61,62,63,64,65,66,67]. Leisure obstacles consist of 12 questions. The leisure obstacles aspect had a KMO value > 0.923, Bartlett’s approximate χ^2^ value of 1849.134, degree of freedom (df) of 66, and significance of 0.000 (*p* < 0.001), making it suitable for factor analysis. The explained variances of the scale were 26.21%, 17.75%, and 16.91%, for a total explained variance of 60.87%. Considering the understanding of the actual state of economic development, all questions were retained after factor analysis. The following three areas were designated: personal obstacles (4 questions), interpersonal obstacles (4 questions), and structural obstacles (4 questions), with 0.922, 0.917, and 0.920, respectively.

For the job satisfaction aspect, the explained variances of the scale were 34.17% and 26.88%, with a total explained variance of 61.05%. Considering the understanding of the actual state of economic development, all these were retained after factor analysis. The following three areas were designated: internal satisfaction factor (4 questions) and external satisfaction factor (4 questions), with 0.902 and 0.899, respectively.

As mentioned above, the contents of the revised final questionnaire on leisure constraints and job satisfaction of middle-aged and elderly health care workers are shown in Table 1.

After collecting the questionnaires and deleting the invalid questionnaires, the study used SPSS 20.0 to establish a document and conduct statistical verification and analysis on the questionnaire. Next, AMOS 23.0 was used to conduct the analysis on the relationship between variables and verification of the plausibility of the research model. As shown in Figure 2.

### 3.3. Data Analysis

Next, after obtaining 260 responses to the formal questionnaire, the researcher used SPSS 22.0 and AMOS 23.0 statistical software to examine the current status of leisure constraints and job satisfaction using basic statistical tests. The structural equation model was used to analyze the correlation between leisure constraints and job satisfaction, and the overall model suitability was determined by examining the offending estimates first, and was based on the χ^2^ test, χ^2^/degree of freedom (DF), goodness-of-fit index (GFI), adjusted goodness-of-fit index (AGFI), root mean square error of approximation (RMSEA), comparative fit index (CFI), and Parsimonious comparative fit index, PCFI).

#### 3.3.1. Offending Estimate

Before undertaking the checking on overall goodness of fit, there needs to be a check on offending estimate; therefore, this study is in compliance with no offending estimate [83,84,85,86,87,88,89]. Offending estimates are used to check whether the estimated coefficients are within an acceptable range before assessing model fitness [90]. Offending estimates exist when the estimate coefficients show 1. negative error variance, 2. insignificant error variance, 3. standardized regression coefficients that are above or too close to 1 (with a threshold of 0.95), and 4. a too large standard error [91,92].

From Table 2 and Table 3, the variance in the research was 0.07 to 0.12 and standardized coefficient was 0.68 to 0.82, not over the standardized value of 0.95 and can thus be used to conduct goodness-of-fit check with the overall model of the study.

#### 3.3.2. Measurement Mode Analysis

The questionnaire of the study was followed by confirmatory factor analysis to verify the reliability and validity, and the project was revised according to the revised indicator modification index (MI) [93]. This study deletes items such as L4 and L10 in the leisure hindrance model and S8 in the job satisfaction model. The rest are within compliance with the standards, so the project remains.

#### 3.3.3. Verification of Convergent Validity

Bagozzi and Yi [36] believe that convergent validity can be derived from the composite reliability (C.R.) and average variance extracted (AVE) of factor perspective. The recommended C.R. value should be greater than 0.7 and AVE greater than 0.5 to show that the questionnaire has convergent validity [90]. The convergent validity of the factors of leisure constraints and job satisfaction was verified, and all factor loadings were in the range of 0.73–0.83, where C.R. values were in the range of 0.77–0.84 and AVE in the range of 0.53–0.64, which is within the normal range [87]. Therefore, this study possessed convergent validity [93], as shown in Table 4.

#### 3.3.4. Discriminant Validity Verification

Long [94] pointed out that discriminant validity is to verify the existence of correlation and significant difference between two different perspectives. The bootstrap 95% confidence interval suggested by Torkzadeh et al. [95,96] was used to check the related coefficient between perspectives. If 1 does not appear, it means it completely correlated and has discriminant validity. From Table 5 and Table 6, the bootstrap 95% confidence intervals are all less than 1, indicating that the research complies with discriminant validity [97,98,99]. 

The present study used bootstrap method to establish the confidence intervals of Pearson’s correlation coefficients between perspectives. If the confidence interval does not contain 1, then the null hypothesis is rejected, i.e., it is perfectly correlated, which means that there is discriminant validity between perspectives [93]. In this verification, the bootstrap assessment was repeated 1,000 times and at the 95% confidence level, it was found that no confidence interval including 1 occurred between perspectives. Therefore, the judgment of the first-order model in the research has discriminative validity.

#### 3.3.5. Analytical Method

The study adopted a mixed-method approach by collecting questionnaires and then using SPSS 22.0 and AMOS 23.0 statistical software to conduct basic statistical tests, *t*-tests, ANOVA tests, Pearson product moment correlation analysis, and related tests. Next, five active health care workers with at least 10 years of experience or academics with medical, public health, or human resource management expertise were interviewed and invited to provide insights into the data results. Finally, after compiling, comparing, and analyzing the data from multiple sources, a multivariate validation analysis was used to examine this issue. Are shown in Table 7.

### 3.4. Ethical Considerations

This research was conducted using the intentional sampling method to identify the subjects, combined with the snowball sampling method. However, the sample collection process needs to be properly and carefully planned, and the difficulties and obstacles in the sample collection process need to be explained [100]. The research is based on the code of ethics [101,102], and sample information is collected. Therefore, all respondents agreed to provide relevant data and understood the main purpose of the study. All questionnaires and interviews were recorded and collected from anonymous and informed respondents.

## 4. Results

### 4.1. Leisure Barriers and Job Satisfaction—Structural Model Analysis

The analysis revealed that leisure constraints had a significantly low correlation (*p* < 0.01) with job satisfaction (0.33), and there were also significant effects of various components. However, among leisure obstacles, interpersonal obstacles (0.942) are the highest, and among job satisfaction, external satisfaction (0.945) is the highest. In terms of research inferences, interpersonal obstacles (0.942) were the highest factor influencing the leisure constraints of middle-aged and elderly health care workers, and the external satisfaction factor (0.945) was the key factor influencing job satisfaction. Among the lei-sure constraints, interpersonal obstacles had the greatest effect on job satisfaction (0.33), and external factors of job satisfaction were crucial to leisure constraints. It was then found that, among the external satisfaction factors of job satisfaction, obtaining a sense of accomplishment (0.377) had the greatest effect on leisure constraints, whereas friends without physical strength (0.352) was the main factor affecting job satisfaction among the interpersonal obstacles of leisure constraints, as shown in Table 8.

### 4.2. Discuss

#### 4.2.1. Discussion of Sample Background

Although the demand and participation of male health care workers have increased with the global promotion of equal work rights for both genders, women still constitute the majority of health care workers in primary care because of the highly technical nature and the delicate work content of professional nursing, general care and medical cleaning. However, although primary health care jobs are stable and the salary grows with seniority, the long working hours and labor-intensive nature of health care, coupled with a workplace environment affected by existing work and epidemic prevention pressures and few opportunities for high-level jobs, make it difficult for new parents or families with childcare needs to take care of their children and afford additional childcare and education costs, and to attract high-level talent. As a result, the nursing workforce is still dominated by women, and highly educated individuals possessing workplace competitiveness, as well as those under the age of 49 who face family pressure and are unable to cope with work pressure, are less inclined to choose a healthcare career.

By hiring sufficient manpower, setting regular duty hours, reducing occupational stress, and offering higher benefits and wages, the willingness of senior medical workers under the age of 49 to seek employment could be improved.

#### 4.2.2. Discussing the Current Situation of Leisure Obstacles and Job Satisfaction among Middle-Aged and Senior Medical Staff

Middle- and high-aged health care workers have already reached the age of marriage and have to spend time with their families and take care of their health while working. Those who are single or in other marital statuses are under stress due to the long hours of clinical work, time difference and poor sleep quality caused by shift work, the broad scope of their jobs, and the high level of medical responsibilities. Therefore, most middle-aged and elderly health care workers were discouraged from engaging in leisure activities, due to personal factors such as a lack of family support and poor health (4.5), a lack of exercise partners (4.57), and insufficient additional exercise budget (4.62).

The advancement structure of the medical employment system is clear, the more years of experience, the greater the chance of advancement. In addition, health care workers are respected by the public for their role in saving lives and taking care of people’s health. However, due to the rising COVID-19 epidemic, the prevalence of the variant virus, and the differences in the awareness and cooperation of individual diagnosed patients, the epidemic has not yet been controlled and there are localized cases of institutional infections, which undermine the confidence of most health care workers [24,25,26]. Therefore, most middle-aged and elderly health care workers believed that good promotion opportunities (4.99) and receiving praise (4.97) make it easy to achieve higher job satisfaction, while dissatisfaction with the current working environment and facilities (4.73), due to the persistence and spread of the epidemic and negligent decision making, prevented them from achieving personal fulfillment in their health care work (4.83).

The advancement system of Taiwan’s medical employment system is well established, and the longer the years of experience, the greater the opportunities, but there is still a shortage of employees in the field. Although the educational requirements for entry into the primary care workforce are low and there are many job opportunities, the life quality and the physical and mental health of middle-aged and elderly healthcare workers are affected by factors such as limited high-level jobs, the highly technical, complex, and laborious nature of the workplace, the shift work mechanism, the fixed and confined workspace, the frequent contact with patients, as well as the current outbreak of epidemics and cluster infection cases in hospital institutions [18], which further impact their willingness to work [19]. Those with college degrees valued environment and facilities planning more, while those who were unmarried or married valued promotion opportunities. Those who were married were more likely to receive appreciation. Those with college and university degrees were more likely to obtain a sense of accomplishment.

By improving the working environment, replenishing departments that lack manpower, stabilizing the duty mechanism for medical staff, improving benefits and wages, planning in-hospital fitness facilities for staff, providing psychological counseling services, developing proper epidemic prevention and medical management strategies, and motivating primary medical staff, the perceptions of middle-aged and elderly medical workers regarding leisure constraints and job satisfaction could be improved, and the feelings of unmarried and highly educated medical professionals about their current workplace and their willingness to plan personal leisure activities could be reversed.

#### 4.2.3. Discussion on the Relationship between Leisure Obstacles and Job Satisfaction

Leisure activities help to improve physical and mental health, and enhance personal performance and productivity [38,67]. However, exercise requires companionship, and effective companionship can promote leisure motivation [98]. In particular, health care workers have monotonous and stressful jobs, and middle-aged and elderly people have to take care of the quality of family life and children’s health, so they have little energy to engage in leisure activities. In addition, it is not easy to find partners for leisure activities because of the similar nature of work in the circle of friends. As a result, it is impossible to properly relieve physical and mental stress, improve work efficiency, or promote work performance and satisfaction. Thus, although there was a low correlation between leisure constraints and job satisfaction (0.33), interpersonal obstacles became the main factor influencing job satisfaction, and friends’ lack of physical strength (0.352) was the main factor preventing middle-aged and elderly healthcare workers from engaging in leisure activities.

Leisure activities can improve physical and mental health, enhance personal performance [38], increase productivity [67], and maintain the individual’s willingness to engage in work or leisure [37,38,39,40]. However, for middle-aged and elderly health care workers, it is more important to be paid for their work and to maintain a stable family and personal lifestyle and daily needs. Particularly in a severe epidemic environment, health care workers can gain job security and encouragement from the public if they can perform well in their jobs, which in turn increases their sense of self-worth; both help stabilize performance and relieve stress for health care workers, freeing up time for planning leisure sports. Thus, job satisfaction has a low level of influence on the willingness of middle-aged and elderly health care workers to engage in leisure, but external job satisfaction level factors have a higher influence on leisure constraints, and the factor of having a sense of accomplishment at work (0.33) is the most critical.

The government, organization directors, or the public could motivate health care workers promptly, improve the welfare of health care workers, design or cooperate with fitness centers nearby, and plan to provide public transportation for staff to engage in leisure activities and children’s pick-ups and drop-offs, so that individuals can plan health maintenance or leisure activities of appropriate intensity for a short period of time to maintain their physical and mental health. In this way, we can enhance the willingness of middle-aged and elderly medical workers for leisure activities, maintain their physical and mental health, improve their job satisfaction, and uphold the quality of medical care.

#### 4.2.4. For Research Objects and Clinical Verification

Health care technology is closely related to human life. Although the standard of employment for primary care jobs only requires a junior college degree, the job is not easily replaced due to its technical and professional content. Compared to other jobs, the pay is good enough to cover basic family expenses, although there are some laborious tasks such as lifting and moving patients and beds. Medical interaction requires the patience, care, and gentleness that most women possess. This makes health care a sacred, stable, and rewarding career.

Due to the design and limitations of the study, however, the data in this study were mainly obtained by analyzing the subjects’ perceptions of the issues from the questionnaire. In the future, if volunteers can be obtained to conduct actual leisure exercise experiments and then continue to analyze their job satisfaction perceptions, the results can further verify our findings.

## 5. Conclusions

Middle-aged and elderly health care workers are faced with existing medical work pressure, COVID-19 epidemic surveillance responsibilities, and pressure from the public, resulting in a less friendly medical work environment in the field. In addition, the unstable duty hours, inadequate benefits and wages, limited time for leisure and exercise, and few choices of venues are the key factors that prevent most middle-aged and elderly health care workers from engaging in leisure activities, thus lowering their job satisfaction. If the working environment can be improved, the manpower can be replenished, the shift mechanism can be stabilized, benefits can be improved, fitness facilities can be planned, psychological counseling services can be provided, and management strategies can be developed to motivate employees, the perception of middle-aged and elderly medical workers about leisure constraints and job satisfaction can be improved, and the unmarried and highly educated medical professionals’ perceptions of their current workplaces and their willingness to plan their leisure activities can be reversed.

The analysis shows that women continued to be the dominant practitioners among middle-aged and elderly health care workers, although the proportion of men was beginning to increase. Although there was a significant correlation between leisure constraints and job satisfaction, only external job satisfaction was significantly associated with leisure constraints and no other significant effects were found, which is inconsistent with the previous literature [23,24,25,35,36]. We believe that the demand for male workers will continue to rise, due to the need to move medical equipment, patients, etc. Focusing on the feelings and attitudes of male workers in the field and improving their willingness to work will be a key to improving the quality of services in healthcare facilities. Furthermore, although leisure constraints are significantly associated with job satisfaction, not all work-level stressors are key contributors to leisure constraints. Therefore, it is suggested that future studies could target male health care workers to understand their feelings and attitudes in the workplace, and analyze the critical influences on external job satisfaction, which would help to fill the research gap.

Based on the above content, the following research suggestions are put forward:The employment willingness of senior medical staff should be valued, and benefits should be improved;Comprehensive on-site medical and epidemic prevention management decisions should be formulated in order to safeguard the existing medical environment in advance;A fitness center should be provided in the facility or through cooperation with a nearby one;Public transportation should be provided for employees to engage in leisure activities and transportation of children to and from their families;Middle-aged and elderly health care workers should schedule 30–60 min of moderate-intensity exercise after work, at least three times a week, to maintain physical and mental health;The medical staff should be supplemented, and the shift schedule mechanism should be stabilized;Find leisure activity partners from work, or plan short-time, moderate-intensity leisure exercises, and engage in leisure regularly to maintain physical and mental health;Exploring the feelings of male health care workers and analyzing the factors that influence external job satisfaction in depth will help fill the research gap;Due to the sampling method and limitations of the study, errors may occur. It is recommended that follow-up researchers use additional sampling methods to obtain a larger amount of data for analysis, and include other age groups in the study, which may complement our findings.

## Figures and Tables

**Figure 1 healthcare-09-00713-f001:**
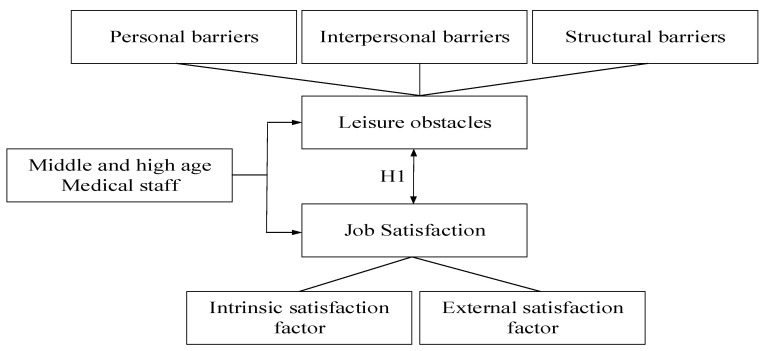
The research structure.

**Figure 2 healthcare-09-00713-f002:**
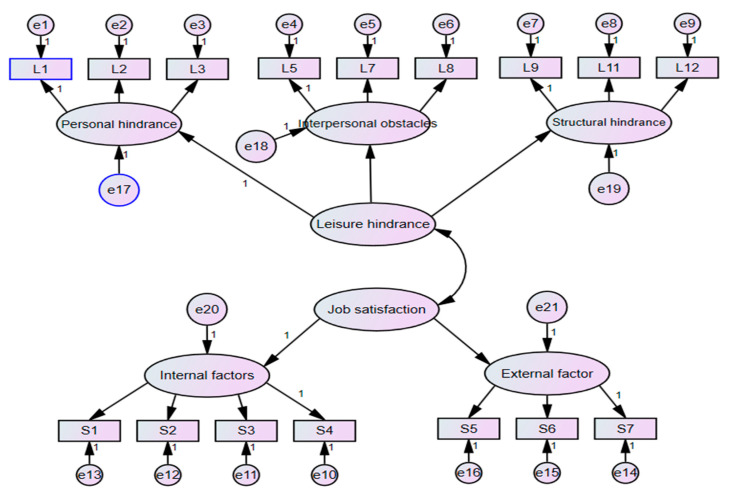
Structure diagram of confirmatory factor analysis of leisure obstacles and job satisfaction.

**Table 1 healthcare-09-00713-t001:** Questionnaire tool description for leisure constraints and job satisfaction of middle-aged and elderly health care workers.

Issue	Content
Background	Gender (male/female); age (45–49/50–54/55–59/60–64); Education (junior college/university/institute); marriage (unmarried/married/other)
leisure constraints	Introverted personality Family does not support Poor health Too many skills Friends have no time Friends have no money Friends have no transportation Friend has no energy I have no money The place is too narrow I have no information Own no transportation
job satisfaction	Environment and facilities Salary Promotion opportunities Get praise Realization ability Get a sense of accomplishment Competent job Profession has special meaning

**Table 2 healthcare-09-00713-t002:** Leisure hindrance scale offending estimate check table.

Item Code		Standardized Regression Coefficient	Deviation Variance
L1	Introverted personality	0.70	0.12
L2	Family does not support	0.74	0.08
L3	Poor health	0.77	0.09
L4	Too many skills	0.69	0.09
L5	Friends have no time	0.79	0.07
L6	Friends have no money	0.68	0.09
L7	Friends have no transportation	0.79	0.07
L8	Friend has no energy	0.80	0.07
L9	I have no money	0.69	0.09
L10	The place is too narrow	0.76	0.08
L11	I have no information	0.75	0.08
L12	Own no transportation	0.68	0.09

**Table 3 healthcare-09-00713-t003:** Job Satisfaction scale offending estimate check table.

Item Code		Standardized Regression Coefficient	Deviation Variance
S1	Environment and facilities	0.75	0.09
S2	Salary	0.73	0.09
S3	Promotion opportunities	0.77	0.09
S4	Get praise	0.77	0.09
S5	Realization ability	0.78	0.08
S6	Get a sense of accomplishment	0.82	0.07
S7	Competent job	0.80	0.08
S8	Profession has special meaning	0.74	0.09

**Table 4 healthcare-09-00713-t004:** Leisure obstacles and job satisfaction model—confirmatory analysis.

Perspective	Index	Standardized Factor Loading	Non-Standardized Factor Loading	S.E.	C.R. (*t*-Value)	*p*	SMC (R^2^)	C.R.	AVE
Personal obstacles	L1	0.73	1.00				0.54	0.80	0.58
	L2	0.78	0.87	0.08	11.50	***	0.60		
	L3	0.78	0.97	0.08	11.43	***	0.62		
Interpersonal obstacles	L5	0.79	1.00				0.62	0.84	0.64
	L7	0.79	0.98	0.07	13.22	***	0.63		
	L8	0.82	1.07	0.07	14.23	***	0.67		
Structural obstacles	L9	0.70	1.00				0.49	0.77	0.53
	L11	0.78	1.10	0.10	10.81	***	0.61		
	L12	0.71	0.99	0.10	10.33	***	0.51		
Intrinsic satisfaction factor	S1	0.76	1.00				0.57	0.84	0.57
	S2	0.73	0.92	0.08	11.73	***	0.53		
	S3	0.77	1.05	0.09	12.13	***	0.59		
	S4	0.77	1.02	0.08	12.10	***	0.59		
External satisfaction factor	S5	0.80	1.00				0.64	0.84	0.64
	S6	0.83	1.04	0.07	14.30	***	0.69		
	S7	0.77	1.02	0.08	12.96	***	0.59		

*** *p* < 0.001.

**Table 5 healthcare-09-00713-t005:** Leisure hindrance—bootstrap 95% confidence interval table of related coefficients.

Parameter			Estimate	Bias-Corrected		Percentile Method	
				Lower Boundary	Upper Boundary	Lower Boundary	Upper Boundary
Personal obstacles	↔	Interpersonal obstacles	0.84	0.75	0.91	0.75	0.91
Personal obstacles	↔	Structural obstacles	0.80	0.69	0.88	0.69	0.89
Interpersonal obstacles	↔	Structural obstacles	0.92	0.85	0.98	0.85	0.99

**Table 6 healthcare-09-00713-t006:** Job satisfaction—bootstrap 95% confidence interval table of related coefficients.

Parameter			Estimate	Bias-Corrected		Percentile Method	
				Lower Boundary	Upper Boundary	Lower Boundary	Upper Boundary
Intrinsic satisfaction factor	↔	External satisfaction factor	0.93	0.85	0.99	0.85	0.99

**Table 7 healthcare-09-00713-t007:** Background of the interviewees and content of the interview.

Identity	Professor(A)	Teacher (B)	Doctors (C)	Nurse (D)	Pharmacy (E)
Seniority	12	20	18	13	20
Interview content	What are the main obstacles to leisure for healthcare workers? Please briefly explain the reasons. What are the work indicators that medical workers care about most when they are engaged in work? Please briefly explain the reasons. Based on the research results of leisure obstacles, that is, job satisfaction, what do you think are the influencing factors?				

**Table 8 healthcare-09-00713-t008:** Correlation analysis of leisure obstacles and job satisfaction.

	Intrinsic Satisfaction Factor		External Satisfaction Factor	Overall Dimensions of Job Satisfaction		Leisure Hinders the Overall Dimension
Personal obstacles	0.226 **		0.311 **	0.285 **		0.899 **
Interpersonal obstacles	0.295 **		0.327 **	0.330 **		0.942 **
Structural obstacles	0.261 **		0.292 **	0.294 **		0.913 **
Intrinsic satisfaction factor	1		0.778 **	0.941 **		0.284 **
External satisfaction factor	0.778 **		1	0.945 **		0.338 **
Overall dimensions of job satisfaction	0.941 **		0.945 **	1		0.330 **
	Leisure obstacles—interpersonal obstacles			Job satisfaction—external factors		
	Friends have no time	Friends have no transportation	Friend has no energy	Ability to achieve	Get a sense of accomplishment	Competent job
Overall dimensions of job satisfaction	0.276 **	0.251 **	0.352 **	-		
Leisure hinders the overall dimension	-			0.305 **	0.337 **	0.284 **

** *p* < 0.01.

## Data Availability

No data support.

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
