# Peer review of "A Study of Leisure Constraints and Job Satisfaction of Middle-Aged and Elderly Health Care Workers in COVID-19 Environment"

_healthcare, 2021, doi:10.3390/healthcare9060713_

Round 1

Reviewer 1 Report

This is an important study. Seen from an occupational medical health point of view the reviewer would like the authors to add in the discussion some kind of limits that we don’t know how many hours this person is really worked. Per week.
Another thing the reviewer like to comment that access to see the questionnaires used 

The authors mention the limited time to have leisure sports activities with no quantitative results. The authors recommendations could include that there should be sufficient time to do that not only at the end of the workday but in the middle of the workdays.

Author Response

Reviewer 1

This is an important study. Seen from an occupational medical health point of view the reviewer would like the authors to add in the discussion some kind of limits that we don’t know how many hours this person is really worked. Per week.

Dear reviewer

We have made your correction. Line 55-58 and line 104-110.

Another thing the reviewer like to comment that access to see the questionnaires used 

Dear reviewer

We have made your correction. Table 1.

The authors mention the limited time to have leisure sports activities with no quantitative results. The authors recommendations could include that there should be sufficient time to do that not only at the end of the workday but in the middle of the workdays.

Dear reviewer

We have made your correction. Line 554-556.

Reviewer 2 Report

Dear author
In the COVID-19 environment, medical workers play a very important role.

The manuscript explores the current situation of medical workers and helps to understand the physical and mental health of the subject and the effectiveness of epidemic prevention.

  1. The abstract fully explains the manuscript theme and results.
  2. There is a large amount of literature in the manuscript to support the results of the inference.
  3. The analytical method conducts data verification in a rigorous manner. 
  4. However, in Table 6, should the title be revised to "Interviewees and topic introduction" 

If the author can confirm the question on this title, it will be a good research topic. 

Author Response

Reviewer 2

However, in Table 6, should the title be revised to "Interviewees and topic introduction"

Dear reviewer

We have made your correction.

Table 7. Background of the interviewees and content of the interview.

Reviewer 3 Report

  1. The research gap should explain in the introduction. Why you chose leisure constraints and job satisfaction and job satisfaction in your model is not clear.
  2. Which theory you use to explain your research model is not clear.

  3. During hypotheses development, you need to combine the theory and your argument, then develop the hypotheses. On page 5, you need to develop argument based on some references.
  4. The theoretical contribution should address in the conclusion.

  5. I suggest that you adopt appropriate academic expression. In general, the writing of the paper could be improved.

  6. Section 4.1 and 4.2 should be deleted.

Author Response

Reviewer 3

The research gap should explain in the introduction. Why you chose leisure constraints and job satisfaction and job satisfaction in your model is not clear.

Dear reviewer

We have made your correction. Line 104-108.

Which theory you use to explain your research model is not clear.

Dear reviewer

We have made your correction. Line 198-210.

During hypotheses development, you need to combine the theory and your argument, then develop the hypotheses. On page 5, you need to develop argument based on some references.

Dear reviewer

We have made your correction. Line 222-230.

The theoretical contribution should address in the conclusion.

Dear reviewer

We have made your correction. Line 505-518.

I suggest that you adopt appropriate academic expression. In general, the writing of the paper could be improved.

Dear reviewer

We have made your correction. Line 222-230.

Section 4.1 and 4.2 should be deleted.

Dear reviewer

We have made your correction.

Round 2

Reviewer 3 Report

  1. Some grammar mistake should revise before submit.
  2. I think SEM theory is not theoretical theory of this paper and you need to use another one to explain your research model.
  3. What is your theoretical contribution, especially for the existing literature is still not clear. 

Author Response

1.Some grammar mistake should revise before submit.
Dear reviewer
We have adjusted some of the manuscript narratives.
Such as lines 138-145. 
2.I think SEM theory is not theoretical theory of this paper and you need to use another one to explain your research model.
Dear reviewer
We have adjusted some of the manuscript narratives.
Such as lines 170-225. 
3.What is your theoretical contribution, especially for the existing literature is still not clear. 
We have adjusted some of the manuscript narratives.
Such as lines 194-207.